# THE MUTUAL INFORMATION UNCERTAINTY RANGE: A NON-PARAMETRIC TEST FOR DEPENDENT CENSORING

## ABSTRACT

Learning a survival prediction model can be viewed as regression with the added complication of censoring. Each subject $x_i$ has a true event time $E_i$ and a censoring time $C_i$, yet we only observe $T_i = \min(E_i, C_i)$ and $\delta_i = \mathbf{1}(E_i \leq C_i)$. Many standard survival methods implicitly assume the $E$ and $C$ are independent, conditioned on $\mathbf{X}$: $E \perp C \mid \mathbf{X}$, which is not always true. To produce effective survival models, it would be useful to know this (in)dependency; however, this is difficult to determine as, for each subject, we observe either $E_i$ xor $C_i$, but never both. To address this challenge, we introduce, for each $t > 0$, indicator variables $E_{i,t} = \mathbf{1}(E_i > t) \in \{0, 1, ?\}$, where "?" represents unobserved values due to censoring; with a similar definition for $C_{i,t}$. We use this set of $\{E_{i,t}, C_{i,t}\}$ over the set of instance $i$ and various times $t$, to develop a nonparametric diagnostic for testing whether $E \perp C \mid \mathbf{X}$, based on the width of the Conditional Mutual Information (CMI) uncertainty range between $E_{i,t}$ and $C_{i,,t}$ given $\mathbf{X}_i$ over the unknown values "?", defined as $\Delta I_t = I_{\max}^t - I_{\min}^t$. Under independence, $\Delta I_t$ follows a characteristic null distribution from random data completions. Dependent censoring imposes structure, producing atypical $\Delta I_t$ values. To make this computation feasible, we formulate the CMI bound computation as a decomposable integer program, which we solve exactly with a dynamic programming algorithm of polynomial complexity. Combined with a permutation test, this yields a scalable, assumption-free tool for detecting dependent censoring. To evaluate the performance of the proposed method, we conducted experiments on synthetic data where both the presence and strength of dependence could be controlled.

## 1 INTRODUCTION

Survival analysis provides a powerful framework for modeling time-to-event data across disciplines, from estimating patient prognosis in clinical trials to predicting component failure in engineering (Clark et al., 2003). A central challenge in this domain is handling right-censored data, where the event of interest is not observed for all subjects due to study termination or loss to follow-up. Standard survival models, including the ubiquitous Cox Proportional Hazards model (Cox, 1972), and other modern models – such as MTLR (Yu et al., 2011), and Deepsurv (Katzman et al., 2018) – rely on the critical assumption of *independent censoring*: conditional on a set of covariates $\mathbf{X}$, the true event time $E$ is independent of the true censoring time $C$, formally $E \perp C \mid \mathbf{X}$.

The independent-censoring assumption means censoring times carry no additional information about event risk given the observed covariates. In practice, this can be violated: for example, in a cancer trial, patients in worsening health may drop out to seek alternative treatment, creating a dependence between early censoring and high risk of death. Such *dependent* censoring can severely bias model estimates, often making treatments appear more effective than they truly are (Lillelund et al., 2025; Gharari et al., 2023).

Despite its importance, it is very difficult to determine if the independent censoring assumption holds in a given dataset. The fundamental difficulty is that, for any given subject, we only observe the follow-up time $T = \min(E, C)$ and an event indicator $\delta = \mathbf{1}\{E \leq C\}$. It means the latent variables

$E$ and $C$ are never simultaneously observed, making their conditional relationship impossible to assess directly (Saïd et al., 2003).

## 1.1 Existing Approaches and Our Contributions

Existing methods have sought to address this question, $E \perp C \mid \mathbf{X}$?, from different perspectives.

Classical approaches rely on **parametric or semi-parametric models**. Tests based on the Cox model, for instance, are conditional on the proportional hazards assumption (Lee & Wolfe, 1998). More flexible joint modeling frameworks, such as **shared frailty models** or **copulas**, directly parameterize the dependence but require strong, untestable assumptions about marginal distributions and the functional form of the dependence, risking misspecification (Rizopoulos, 2012). **Non-parametric tests** relax these distributional assumptions (Sun & Lee, 2011). Rank-based procedures can detect monotone dependence but miss more complex relationships (Lee & Wolfe, 1995), (Flandre, 2022). Another approach, **sensitivity analysis**, aims to bound the impact of potential dependence rather than test for its presence, serving as a tool for robust estimation rather than hypothesis testing (Scharfstein et al., 1999).

This review highlights a critical gap in the literature: a lack of methods that are simultaneously (1) non-parametric, (2) capable of detecting general forms of dependence, (3) specifically designed for the censored data structure, and (4) framed as a principled hypothesis test.

**Our Contributions.** This paper introduces a novel framework to fill this gap by reframing the problem: instead of measuring a single dependence value, we measure the structural constraints that dependence imposes on the data. Our key contributions are:

1. **A Theoretically Grounded Discretization.** We introduce a discrete-time model, for eahch $t > 0$, with indicator variables $E_t = \mathbf{1}(E > t)$ and $C_t = \mathbf{1}(C > t)$ and prove (Proposition 1) that the continuous-time independence assumption ($E \perp C \mid \mathbf{X}$) is a necessary condition for independence between discrete-time survival and censoring indicators ($E_t \perp C_t \mid \mathbf{X}$). This result provides the theoretical license to use discrete information-theoretic tools for inference dependence on the continuous process.

2. **A Novel Information-Theoretic Test Statistic ($\Delta I$).** We define our test statistic as the width of the Conditional Mutual Information (CMI) uncertainty range. This measures how dependence constrains the space of possible data completions. An outlier $\Delta I$ value (unusually narrow or wide) relative to its null distribution provides evidence against independence.

3. **A General and Robust Test.** Our use of CMI allows the detection of *any* form of statistical dependence, making it more powerful than rank-based tests (Lee & Wolfe, 1995). By embedding this in a stratified permutation framework, our test is non-parametric, avoiding the strong assumptions of model-based approaches.

4. **A Computationally Tractable Solution.** We overcome the combinatorial challenge of optimizing CMI over all imputations by formulating it as a decomposable integer program. We solve this exactly and efficiently using a dynamic programming algorithm, making our theoretically grounded test a practical diagnostic tool.

Collectively, these contributions provide the first assumption-free, statistically rigorous, and computationally feasible framework for directly testing the independent censoring assumption in its full generality.

## 2 Background: Survival Analysis and Independent Censoring

Survival analysis models the time until an event occurs. The primary challenge is that the true event time is not always observed due to right-censoring.

**Definition 1 (The Time-to-Event Setting)** *Let $E$ and $C$ be latent, non-negative random variables for the true event and censoring times for an instance, respectively. The observed data for each instance is a tuple $(T, \delta, \mathbf{X})$, where $\mathbf{X}$ is a vector of covariates, and $T = \min(E, C)$ and $\delta = \mathbf{1}\{E \leq C\}$ are observed time and event indicator, respectively. The indicator $\delta = 1$ signifies an observed event, while $\delta = 0$ indicates the observation was **right-censored**.*

The distribution of the event time $E$ is typically characterized by one of two mathematically-equivalent functions.

**Definition 2 (Survival and Hazard Functions)** *Given covariates* $\mathbf{X}$, *the event process is described by:*

- *The **Survival Function**,* $S(t|\mathbf{X}) = P(E > t|\mathbf{X})$, *which is the probability that the event occurs after time* $t$.

- *The **Hazard Function**,* $\lambda(t|\mathbf{X}) = \lim_{\Delta t \to 0} \frac{P(t \leq E < t + \Delta t | E \geq t, \mathbf{X})}{\Delta t}$, *which represents the instantaneous rate of event occurrence at time* $t$, *given survival up to* $t$.

Canonical survival models, from the Kaplan-Meier estimator to the Cox proportional hazards model (Cox, 1972), rely on the following critical, untestable assumption about the censoring mechanism.

**Definition 3 (Independent Censoring)** *The event time* $E$ *and censoring time* $C$ *are **independently censored** if they are conditionally independent given the covariates* $\mathbf{X}$:

$$E \perp C \mid \mathbf{X}. \tag{1}$$

*This assumption posits that, conditional on covariates, the censoring mechanism provides no further information about a subject's risk of the event.*

In practice, this assumption is often violated, leading to **dependent censoring** ($E \not\perp C \mid \mathbf{X}$). For instance, when measuring time until death for patients taking some treatment, a patient whose health deteriorates may drop out of a clinical trial. Here, the censoring event (dropout) is directly correlated with a higher event risk (e.g., mortality). Ignoring such dependence invalidates standard methods and can lead to severely biased conclusions.

## 3 FROM CONTINUOUS PROCESSES TO DISCRETE-TIME REPRESENTATIONS

To test the continuous-time independence hypothesis, $E \perp C \mid \mathbf{X}$, we translate the problem into a discrete-time domain. This strategic shift is dually motivated. First, the continuous-time hypothesis is fundamentally unidentifiable, as the latent event and censoring times are never jointly observed. Second, this discretization enables the use of Conditional Mutual Information (CMI), which is naturally defined and robustly estimated from the empirical counts of discrete variables. By mapping the continuous survival process to a sequence of binary indicators, we can construct the contingency tables necessary to apply our information-theoretic framework.

### 3.1 DISCRETE-TIME REPRESENTATION OF SURVIVAL DATA

To discretize the continuous process, we select an ordered set of time points $\mathcal{T} = \{t_1, t_2, \ldots, t_K\}$, typically the unique event times in the dataset. At each time point $t_k \in \mathcal{T}$, we define two binary random variables for each subject:

- $E_{t_k} := \mathbf{1}\{E > t_k\}$, the **survival status indicator**. $E_{t_k} = 1$ if the subject has survived past time $t_k$.

- $C_{t_k} := \mathbf{1}\{C > t_k\}$, the **censoring status indicator**. $C_{t_k} = 1$ if the subject has not been censored by time $t_k$.

A critical aspect of this representation is the induced missingness. After a subject is censored at time $C$, their true event status $E_t$ for all $t > C$ becomes unobserved. Similarly, once an event occurs at time $E$, the subsequent censoring status $C_t$ for $t > E$ is not observed, as the subject is no longer in the risk set. Table 1 and Fig. 1 illustrate this transformation, showing the discrete-time trajectories for four subjects based on their observed event xor censoring times. As Subject A experiences the event at $t = 1.5$: their $[E_t, C_t]$ transition from $[1, 1]$ before $t = 1.5$, to become $[0, ?]$ afterwards. For Subject C, censored at $t = 3$, the survival status $E_t$ is known to be 1 up to that point but becomes unobserved ? thereafter and $C_t$ goes from 1 to 0.

Table 1: Discretization of continuous survival data. The observed data (T, $\delta$) generate a sequence of discrete-time survival ($E_t$) and censoring ($C_t$) indicators, which depend on the underlying latent continuous times (E, C). "?" denotes values that are unobserved due to a prior event or censoring.

|       | **Observed Data** | | **Latent Times** | | $t=0$ | | $t=1.5$ | | $t=2.1$ | | $t=3.0$ | | $t=5.0$ | |
| Sub.  | $T$ | $\delta$ | **E** | **C** | $E_0$ | $C_0$ | $E_{1.5}$ | $C_{1.5}$ | $E_{2.1}$ | $C_{2.1}$ | $E_3$ | $C_3$ | $E_5$ | $C_5$ |
|-------|-----|----------|-------|-------|-------|-------|-----------|-----------|-----------|-----------|-------|-------|-------|-------|
| A     | 1.5 | 1        | 1.5   | $\geq 1.5$ | 1 | 1 | **0** | ? | **0** | ? | **0** | ? | **0** | ? |
| B     | 2.1 | 1        | 2.1   | $\geq 2.1$ | 1 | 1 | 1     | 1 | **0** | ? | **0** | ? | **0** | ? |
| C     | 3.0 | 0        | $> 3.0$ | 3.0 | 1 | 1 | 1     | 1 | 1     | 1 | ? | **0** | ? | **0** |
| D     | 5.0 | 0        | $> 5.0$ | 5.0 | 1 | 1 | 1     | 1 | 1     | 1 | 1 | 1 | ? | **0** |

Figure 1: Timeline visualization of survival data for four subjects. A single line represents each subject's status at each $t > 0$: 'At Risk' (green: $E_t = 1, C_t = 1$), 'Event Occurred earlier' (red: $E_t = 0$. $C_t =$?), or 'Censored earlier' (yellow: $E_t =$?, $C_t = 0$). An event ($\delta = 1$) is marked with a filled circle, and censoring ($\delta = 0$) is marked with an open circle.

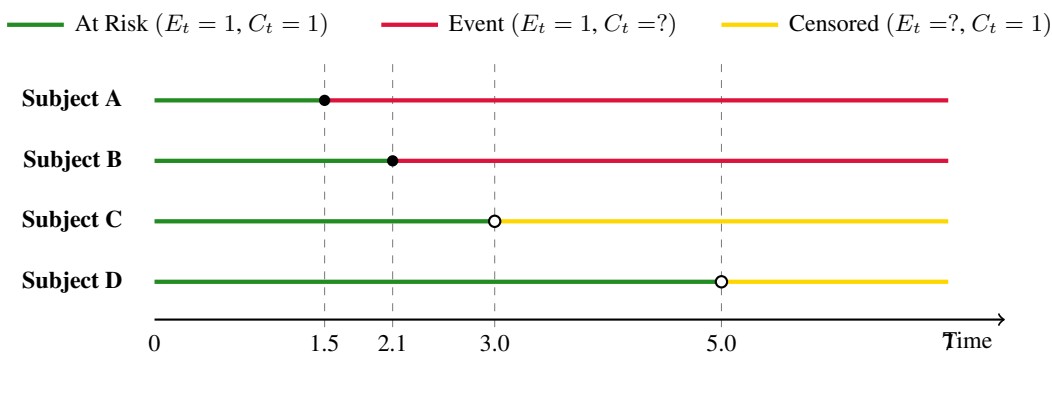

## 3.2 EQUIVALENCE OF CONTINUOUS AND DISCRETE-TIME INDEPENDENT CENSORING

A crucial question is whether the independent censoring assumption from the continuous domain translates to the discrete representation. We demonstrate that it does, which provides the theoretical justification for our testing framework.

**Proposition 1** *Let $E$ and $C$ be continuous event and censoring times, and let $\mathbf{X}$ be a vector of covariates. The continuous-time independent censoring assumption, $H_0^C : E \perp C \mid \mathbf{X}$, implies the discrete-time independent censoring assumption,*

$$\forall t > 0, \quad H_0^D(t) : E_t \perp C_t \mid \mathbf{X}, \tag{2}$$

*where $E_t = \mathbf{1}\{E > t\}$ and $C_t = \mathbf{1}\{C > t\}$.*

This proposition is the cornerstone of our testing methodology. Its contrapositive guarantees that if there exist a $t$ such that $H_0^D(t)$ fails, then $H_0^C$ fails. This insight transforms the testing problem: we need only find a single, statistically significant instance of discrete dependence to reject the global null hypothesis. Accordingly, our global test is designed to find the maximum deviation from independence across a set of discrete time points, while correcting for the multiple comparisons involved in such a search.

## 4 AN INFORMATION-THEORETIC FRAMEWORK FOR DETECTING DEPENDENCE

Building on the equivalence between continuous and discrete-time independence, we now introduce a non-parametric framework to test the hypothesis of independent censoring. Our approach is rooted in information theory and tailored to handle the systemic censoring inherent in survival analysis data.

## 4.1 QUANTIFYING DEPENDENCE WITH CONDITIONAL MUTUAL INFORMATION.

The core of our test is the **CMI**, a fundamental measure of the strength of dependence between two random variables given a third. For a chosen time point $t$, we measure the dependence between the survival status $E_t$ and the censoring status $C_t$ conditioned on the covariates $\mathbf{X}$. The CMI is defined, for any fixed t, as:

$$I(E_t; C_t \mid \mathbf{X}) = \mathbb{E}_{p(E_t, C_t, \mathbf{x})} \left[ \log \frac{p(E_t, C_t \mid \mathbf{x})}{p(E_t \mid \mathbf{x})p(C_t \mid \mathbf{x})} \right]. \tag{3}$$

Intuitively, the CMI quantifies the reduction in uncertainty about a subject's survival status at time $t$ if we know their censoring status, given that we have already accounted for the information in their covariates $\mathbf{X}$. The conditional independence hypothesis $E_t \perp C_t \mid \mathbf{X}$ holds if and only if $I(E_t; C_t \mid \mathbf{X}) = 0$. For discrete covariates $\mathbf{X}$, we can estimate the CMI empirically from a dataset of size $n$. The standard estimator is:

$$\hat{I}(E_t; C_t \mid \mathbf{X}) = \frac{1}{n} \sum_{\mathbf{x} \in \mathcal{X}} \sum_{e_t \in \{0,1\}} \sum_{c_t \in \{0,1\}} N(e_t, c_t, \mathbf{x}) \log \left( \frac{N(\mathbf{x})N(e_t, c_t, \mathbf{x})}{N(e_t, \mathbf{x})N(c_t, \mathbf{x})} \right), \tag{4}$$

where $N(\cdot)$ represents the empirical counts of observations. For instance, $N(e_t, c_t, \mathbf{x})$ is the number of subjects with covariates $\mathbf{x}$ having survival status $e_t$ and censoring status $c_t$ at time $t$, and $N(\mathbf{x})$ is the total number of subjects in that stratum. The stratum means the subgroup of the study population defined by a specific covariate profile $\mathbf{x}$. Formally, a stratum is the equivalence class of individuals sharing the same covariate vector $\mathbf{x}$.

### 4.1.1 CMI BOUNDS VIA OPTIMIZATION OVER FEASIBLE COMPLETIONS

Directly computing the empirical CMI from Equation 4 is impossible due to ambiguity from right-censoring. At any time $t$, the full contingency table of counts is unobserved for two subgroups: (1) subjects censored before $t$ (meaning their survival status $E_t$ is unknown), and (2) subjects who experienced an event before $t$ (meaning their counterfactual censoring status $C_t$ is unknown). Here, in each case, we need to (implicitly) consider both possible assignments to the missing value (either $E_t$ or $C_t$). If a dataset contains $\ell$ and $m$ such subjects, this ambiguity creates a combinatorially infeasible solution space of $2^{\ell+m}$ possible data completions, rendering a brute-force approach intractable.

**Optimization Framework.** We circumvent this by reframing the task as an optimization problem: finding the minimum and maximum CMI values over the space of all valid imputations. For each time $t$, within each covariate stratum $\mathbf{x}$, let $\ell_{\mathbf{x}}$ and $m_{\mathbf{x}}$ be the counts of subjects censored and with an event before time $t$, respectively. We introduce integer decision variables: $u_{\mathbf{x}}^{10} \in \{0, 1, \cdots, \ell_{\mathbf{x}}\}$ represents the number of the $\ell_{\mathbf{x}}$ censored subjects imputed as "alive" ($E_t = 1, C_t = 0$), and $u_{\mathbf{x}}^{01} \in \{0, 1, \cdots, m_{\mathbf{x}}\}$ represents the number of the $m_{\mathbf{x}}$ event subjects imputed as "uncensored" ($E_t = 0, C_t = 1$). The remaining ambiguous subjects are assigned to the "dead and censored" ($E_t = 0, C_t = 0$) category. These variables, along with the fully observed counts $N^{obs}(s, d, \mathbf{x})$, define the completed contingency table:

$$N(\mathbf{x}) = N^{obs}(1, 1, \mathbf{x}) + \ell_{\mathbf{x}} + m_{\mathbf{x}} \quad , \quad N(0, 0, \mathbf{x}) = (\ell_{\mathbf{x}} - u_{\mathbf{x}}^{10}) + (m_{\mathbf{x}} - u_{\mathbf{x}}^{01})$$
$$N(1, 1, \mathbf{x}) = N^{obs}(1, 1, \mathbf{x}) \quad , \quad N(1, 0, \mathbf{x}) = u_{\mathbf{x}}^{10} \quad , \quad N(0, 1, \mathbf{x}) = u_{\mathbf{x}}^{01}.$$

Regarding to empirical definition of CMI (4), the problem of finding the CMI bounds, $I_{\min}^t$ and $I_{\max}^t$, reduces to solving the following nonlinear integer programs:

$$I_{\min}^t = \underset{\{u_{\mathbf{x}}^{10}, u_{\mathbf{x}}^{01}\}_{\mathbf{x} \in \mathcal{X}}}{\text{minimize}} \quad \hat{I}(E_t; C_t \mid \mathbf{X}) \tag{5}$$

$$\text{s.t.} \quad u_{\mathbf{x}}^{10} \in \{0, 1, \cdots, \ell_{\mathbf{x}}\}, \quad u_{\mathbf{x}}^{01} \in \{0, 1, \cdots, m_{\mathbf{x}}\} \quad \forall \mathbf{x} \in \mathcal{X}.$$

where $\hat{I}(E_t; C_t \mid \mathbf{x}) = \sum_{\mathbf{x} \in \mathcal{X}} \hat{I}_{\mathbf{x}}^t$, and

$$\hat{I}_{\mathbf{x}}^t = \frac{1}{n} \left[ u_{\mathbf{x}}^{10} \log\left( \frac{N(\mathbf{x}) \, u_{\mathbf{x}}^{10}}{\left(N^{obs}(1,1,\mathbf{x}) + u_{\mathbf{x}}^{10}\right)\left(\ell_{\mathbf{x}} + m_{\mathbf{x}} - u_{\mathbf{x}}^{01}\right)} \right) \right.$$

$$+ \, u_{\mathbf{x}}^{01} \log\left( \frac{N(\mathbf{x}) \, u_{\mathbf{x}}^{01}}{\left((\ell_{\mathbf{x}} - u_{\mathbf{x}}^{10}) + m_{\mathbf{x}}\right)\left(N^{obs}(1,1,\mathbf{b}) + u_{\mathbf{x}}^{01}\right)} \right)$$

$$+ \, N^{obs}(1,1,\mathbf{x}) \log\left( \frac{N(\mathbf{x}) \, N^{obs}(1,1,\mathbf{x})}{\left(N^{obs}(1,1,\mathbf{x}) + u_{\mathbf{x}}^{10}\right)\left(N^{obs}(1,1,\mathbf{x}) + u_{\mathbf{x}}^{01}\right)} \right)$$

$$+ \, \left((\ell_{\mathbf{x}} - u_{\mathbf{x}}^{10}) + (m_{\mathbf{x}} - u_{\mathbf{x}}^{01})\right) \log\left( \frac{N(\mathbf{x}) \left((\ell_{\mathbf{x}} - u_{\mathbf{x}}^{10}) + (m_{\mathbf{x}} - u_{\mathbf{x}}^{01})\right)}{\left((\ell_{\mathbf{x}} - u_{\mathbf{x}}^{10}) + m_{\mathbf{x}}\right)\left(\ell_{\mathbf{x}} + m_{\mathbf{x}} - u_{\mathbf{x}}^{01}\right)} \right) \left. \right].$$

$I_{\max}^t$ can be calculated in the same way as $I_{\min}^t$, by replacing 'minimize' with 'maximize' in 5 as well.

**Tractable Solution via Separability.** Although each Equation 5 appears complex, it is critically separable: the objective is a sum of stratum-specific terms, and the decision variables for one stratum $\mathbf{x}_i$ are independent of those for any other stratum $\mathbf{x}_j$. This property allows the global optimization to be decomposed into a set of independent subproblems, one for each stratum: $\min \hat{I}_t = \sum_{\mathbf{x} \in \mathcal{X}} \min_{\{u_{\mathbf{x}}^{10}, u_{\mathbf{x}}^{01}\}} \left[ \hat{I}_{\mathbf{x}}^t(u_{\mathbf{x}}^{10}, u_{\mathbf{x}}^{01}) \right]$. For each stratum, this amounts to a small integer program over just two variables. Because $\ell_{\mathbf{x}}$ and $m_{\mathbf{x}}$ are typically small, the optimal imputation can be found efficiently by enumerating all $(\ell_{\mathbf{x}} + 1) \times (m_{\mathbf{x}} + 1)$ points on the integer grid. This stratum-wise optimization is exponentially faster than the naive approach, making the computation of exact CMI bounds feasible even for large datasets.

## 4.2 THE MI UNCERTAINTY RANGE AND PERMUTATION TESTING

Our proposed test statistic is the **MI uncertainty range**, defined as $\Delta I_t = I_{\max}^t - I_{\min}^t$, where

$$I_{\max}^t = \max_{\{u_{\mathbf{x}}^{10}, u_{\mathbf{x}}^{01}\}_{\mathbf{x} \in \mathcal{X}}} \hat{I}(E_t; C_t \mid \mathbf{X}) \quad \text{and} \quad I_{\min}^t = \min_{\{u_{\mathbf{x}}^{10}, u_{\mathbf{x}}^{01}\}_{\mathbf{x} \in \mathcal{X}}} \hat{I}(E_t; C_t \mid \mathbf{X}) \tag{6}$$

for each $t > 0$. The central idea is that the true data-generating process—whether independent or dependent—imposes a specific structure on the data, which in turn constrains the range of possible CMI values achievable through imputation. The nature of this constraint provides the signal for our test.

- **Under Independent Censoring (Null Hypothesis):** The relationship between event and censoring times is unstructured and random (conditional on covariates). This generates a "natural" or "baseline" distribution for the uncertainty range, $\Delta I_t$, which we can characterize empirically via permutation.

- **Under Dependent Censoring (Alternative Hypothesis):** A systematic relationship exists between the event and censoring processes. This underlying structure is a non-random signal that constrains the optimization process, producing a $\Delta I_t$ value that is an outlier relative to the null distribution. This deviation can manifest as an unusually *narrow* range (if the dependence is a strong signal that persists across all imputations) or an unusually *wide* range (if the structure allows for extreme CMI values not achievable under random permutations).

Therefore, a $\Delta I_t$ value that is an outlier in either direction constitutes evidence against the null hypothesis. To determine if an observed range $\Delta I_t^{\text{obs}}$ is statistically significant, we must compare it against a "null world" where we know the independence assumption holds. We construct this null distribution via a carefully designed permutation test.

**Constructing the Null Distribution via Stratified Permutation.** The core question a hypothesis test must answer is: "Could a result like this have occurred by random chance?" To answer this for our observed statistic, $\Delta I_t^{\text{obs}}$, we need to generate an empirical null distribution—the distribution of $\Delta I_t$ values we would expect to see if the null hypothesis, $H_0 : E \perp C \mid \mathbf{X}$, were true.

We simulate this null world via permutation. The key insight is that if censoring and events are truly independent *conditional on covariates* $\mathbf{X}$, then within any stratum of subjects with the same $\mathbf{X}$, the observed outcome (event or censored) is effectively a random label. By randomly shuffling the event indicators, $\delta_i$, among subjects *within each covariate stratum*, we can create new, permuted datasets, but that cannot perfectly simulate the null hypothesis. Therefore, we introduce a null distribution process based on **random survival forest (RSF)** (Ishwaran et al., 2008), which is explained in the next subsection. This procedure has two critical features:

1. **It Breaks Dependence:** The shuffling severs any potential true link between the event and censoring processes that might be driven by a confounder (e.g., disease severity).

2. **It Preserves Covariate Structure:** The stratification is crucial. By shuffling only *within* strata, we preserve the real, observed relationships between the covariates $\mathbf{X}$ and the outcomes. For example, if older patients have more events, our permuted datasets will still show that older patients have more events. We do not shuffle across strata, as this would test the wrong (marginal) hypothesis and lead to an invalid null distribution.

By calculating $\Delta I_t^b$ for thousands of such permuted datasets, we build a robust empirical distribution of the test statistic under the null, providing the correct baseline for assessing the significance of our observed result.

### 4.3 A Non-Parametric Permutation Test for Dependent Censoring

To formally test for the presence of dependent censoring, we propose a robust, non-parametric permutation test sensitive to dependencies that may exist only within specific covariate subgroups or at particular time intervals. The core of our test relies on the $\Delta I_t$ statistic, which is computed for each covariate stratum $s$ at a pre-defined set of time points $\{t_k\}_{k=1}^{K}$. To aggregate evidence, we employ a two-stage procedure. First, we compute a stratum-level test statistic $\Lambda^s$ by taking the maximum $\Delta I_{t_k}^s$ value across all time points. A stratum-specific p-value, $p_s$, is then calculated from its permutation distribution. Second, we combine these p-values from all strata using Fisher's method (Fisher, 1925) to obtain a single, global test statistic $F = -2 \sum_s \log p_s$. This hierarchical approach allows the test to detect both strong, localized dependence in a single stratum and weaker but consistent dependence across multiple strata.

The validity of this test hinges on a robust method for generating datasets under the null hypothesis of conditional independence ($E \perp C \mid \mathbf{X}$). We introduce a powerful, non-parametric method based on the "impute-and-permute" paradigm, which leverages the flexibility of RSF. The objective is to generate null datasets where the event and censoring processes are independent conditional on covariates ($E^* \perp C^* \mid \mathbf{X}$), while faithfully preserving the complex marginal relationships between the covariates and the outcomes ($P(E^*|\mathbf{X}) \approx P(E|\mathbf{X})$ and $P(C^*|\mathbf{X}) \approx P(C|\mathbf{X})$). The procedure is detailed in Algorithm 1.

This null generation process is then embedded within our complete testing procedure, as detailed in Algorithm 2. The final global p-value is determined by comparing the observed Fisher statistic $F_{\text{obs}}$ to its own null distribution, constructed by re-computing $F$ for each permutation. Its primary limitation is the potential for reduced performance in settings with very high-dimensional covariates or sparse data within strata.

## 5 Numerical Experiments on Synthetic Data

We evaluate our proposed test on synthetic data, where the ground truth is known, to assess its statistical power (detecting true dependence) and control of Type I error (avoiding false positives). We utilize four standard generative models based on copula functions: Gaussian (Li, 2000), Frank (Genest, 1986), Clayton (Clayton, 1978), and Gumbel (Gumbel, 1960), which allow precise control over the censoring mechanism, thus ensuring a comprehensive evaluation across different dependence

structures. To investigate the efficiency of the proposed method for detecting dependent censoring induced by a confounding variable, a frailty data generation is applied as well. We consider a synthetic dataset with four binary-value features and 1000 instances. For Algorithm 2, $K = 9$ quintiles $\{0.1, 0.2, 0.3, 0.4, \cdots, 0.8, 0.9\}$ in each dataset are picked as time steps $t_k$, and the number of permutations is set to $B = 200$. For different values of dependence parameters, the $p$-value results for the five data generation methods are reported in Table 2. A $p$-value close to zero indicates dependent censoring. The detailed description of these generative models is provided in the Appendix.

Table 2: P-values across data generation methods with varying parameters using four binary covariates.

| **Gaussian Copula** ($\theta \in (-1, 1)$) | | | | | | | | |
|---|---|---|---|---|---|---|---|---|
| $\theta$ | -0.7 | -0.5 | -0.3 | 0 | 0.3 | 0.35 | 0.5 | 0.7 |
| P-value | 0.005 | 0.005 | 0.25 | 0.96 | 0.14 | 0.04 | 0.005 | 0.005 |
| **Clayton Copula** ($\theta \in [0, \infty)$) | | | | | | | | |
| $\theta$ | 0 | 0.25 | 0.8 | 1 | 2 | 3 | 4 | 5 |
| P-value | 0.85 | 0.61 | 0.018 | 0.025 | 0.005 | 0.005 | 0.005 | 0.005 |
| **Gumbel Copula** ($\theta \in [1, \infty)$) | | | | | | | | |
| $\theta$ | 1 | 1.2 | 1.4 | 1.6 | 2 | 3 | 4 | 5 |
| P-value | 0.95 | 0.31 | 0.01 | 0.005 | 0.005 | 0.005 | 0.005 | 0.005 |
| **Frank Copula** ($\theta \in (-\infty, \infty) \setminus \{0\}$) | | | | | | | | |
| $\theta$ | -1 | -0.5 | -0.1 | 0.1 | 1 | 1.5 | 2 | 3 |
| P-value | 0.005 | 0.005 | 0.54 | 0.96 | 0.41 | 0.04 | 0.005 | 0.005 |
| **Frailty** ($\alpha_E = \alpha_C = \alpha \in [0, \infty)$) | | | | | | | | |
| $\alpha$ | 0 | 1 | 1.5 | 1.7 | 2 | 3 | 4 | 5 |
| P-value | 0.98 | 0.84 | 0.56 | 0.19 | 0.03 | 0.005 | 0.005 | 0.005 |

The permutation testing framework in Section 4.3 assumes discrete covariates $\mathbf{X}$ to define strata for both the permutation scheme and the CMI calculation. Since real datasets often contain continuous variables (e.g., age, blood pressure), we address this by discretizing each continuous feature into bins (e.g., quantiles or domain-informed intervals). The resulting discrete strata both enable stratified permutation of censoring times and make the CMI-based statistic $\Delta I$ tractable, avoiding the challenges of non-parametric estimation with continuous covariates.

Although discretization reduces covariate granularity, it provides a simple and effective solution that balances theoretical soundness with practical usability. The number of bins serves as a key hyperparameter, guided by dataset size and characteristics. To illustrate this idea, we repeated the experiments using four continuous features that were binarized into two bins after data generation, with results reported in Table 3.

## 5.1 DISCUSSION

The experiments on synthetic data confirm that our test reliably controls Type I error and has strong power against dependence across diverse data-generating mechanisms. In both binary and continuous covariate settings, the procedure yields large $p$-values under independence (e.g., $\theta = 0$ for Gaussian/Clayton/Frank, $\theta = 1$ for Gumbel, $\alpha = 0$ for frailty), demonstrating proper error control.

As dependence strengthens, $p$-values quickly approach zero, showing high sensitivity across all copulas and frailty models. For Gaussian copulas, both positive and negative correlations are detected, while for Clayton and Gumbel, dependence is identified at relatively small parameter values. Frank requires stronger dependence before rejection, and frailty shows a gradual decline in $p$-values, re-

Table 3: P-values across data generation methods with varying parameters, using four continuous covariates.

| Gaussian Copula ($\theta \in (-1, 1)$) | | | | | | | |
|---|---|---|---|---|---|---|---|
| $\theta$ | -0.7 | -0.5 | -0.3 | 0 | 0.3 | 0.35 | 0.5 | 0.7 |
| P-value | 0.005 | 0.005 | 0.1 | 0.93 | 0.26 | 0.16 | 0.01 | 0.005 |
| **Clayton Copula** ($\theta \in [0, \infty)$) | | | | | | | |
| $\theta$ | 0 | 0.25 | 0.8 | 1 | 2 | 3 | 4 | 5 |
| P-value | 0.98 | 0.7 | 0.43 | 0.12 | 0.005 | 0.005 | 0.005 | 0.005 |
| **Gumbel Copula** ($\theta \in [1, \infty)$) | | | | | | | |
| $\theta$ | 1 | 1.2 | 1.4 | 1.6 | 2 | 3 | 4 | 5 |
| P-value | 0.89 | 0.35 | 0.03 | 0.005 | 0.005 | 0.005 | 0.005 | 0.005 |
| **Frank Copula** ($\theta \in (-\infty, \infty) \setminus \{0\}$) | | | | | | | |
| $\theta$ | -1 | -0.5 | -0.1 | 0.1 | 1 | 1.5 | 2 | 3 |
| P-value | 0.005 | 0.005 | 0.4 | 0.98 | 0.3 | 0.005 | 0.005 | 0.005 |
| **Frailty** ($\alpha_E = \alpha_C = \alpha \in [0, \infty)$) | | | | | | | |
| $\alpha$ | 0 | 1 | 1.5 | 1.7 | 2 | 3 | 4 | 5 |
| P-value | 0.98 | 0.92 | 0.56 | 0.18 | 0.02 | 0.005 | 0.005 | 0.005 |

flecting latent-factor effects. Results with discretized continuous covariates (Table 3) mirror those with binary covariates (Table 2), confirming that discretization maintains the test's effectiveness.

Overall, the findings indicate that the proposed method achieves robust Type I error control and strong power under a wide variety of dependence structures, supporting its use in practical survival settings with both binary and continuous covariates.

## 6 CONCLUSION

This paper introduced a principled, non-parametric method to test the crucial independent censoring assumption in survival analysis, a foundational yet often unverifiable prerequisite for many models. Our core innovation is to leverage, rather than circumvent, the ambiguity inherent in censored data. We defined a novel test statistic, the **MI uncertainty range** ($\Delta I_t$), based on the bounds of Conditional Mutual Information over all valid data completions. To make this computationally feasible, we developed a tractable algorithm that solves the underlying integer program exactly via a stratum-wise decomposition. This statistic is embedded within a robust permutation test that uses a Random Survival Forest-based "impute-and-permute" scheme to generate a valid null distribution. Extensive experiments on synthetic data, using a wide range of copula and frailty models, confirmed that our method effectively controls Type I error while demonstrating high statistical power to detect diverse dependence structures. The test's strong performance was maintained for both discrete and continuous covariates handled via discretization. In conclusion, we provide practitioners with a scalable and assumption-free diagnostic tool to validate a critical assumption in time-to-event analysis. This work strengthens the credibility of survival models and promotes more robust and reliable scientific findings.

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

# A APPENDIX

## A.1 PROOF OF PROPOSITION 1

A fundamental property of conditional independence is its preservation under measurable transformations. If random variables $U \perp V \mid Z$, then for any measurable functions $g$ and $h$, it follows that $g(U, Z) \perp h(V, Z) \mid Z$ (Durrett, 2019).

In our context, let $U = E$, $V = C$, and $Z = \mathbf{X}$. The indicator functions $g(e) = \mathbf{1}\{e > t\}$ and $h(c) = \mathbf{1}\{c > t\}$ are measurable functions for any fixed $t$. Applying this property, the assumption $E \perp C \mid \mathbf{X}$ directly implies:

$$\mathbf{1}\{E > t\} \perp \mathbf{1}\{C > t\} \mid \mathbf{X} \quad \forall t > 0,$$

which is precisely the statement $H_0^D(t)$. $\qquad\square$

## A.2 RANDOM SURVIVAL FOREST (RSF)

Random Survival Forests (RSF), introduced by Ishwaran et al. (2008), extend Breiman's Random Forests to right-censored survival data. RSF is a fully non-parametric ensemble learning method that constructs multiple survival trees and aggregates them to obtain stable estimates of survival functions.

**Algorithmic Outline.** Each survival tree in RSF is grown on a bootstrap sample of the original data. At each node:

1. A random subset of candidate covariates is selected.
2. The optimal split is chosen to maximize survival difference between daughter nodes, commonly using the log-rank splitting criterion.

The tree is grown until a stopping rule is met (e.g., minimum terminal node size). For each terminal node, the cumulative hazard function (CHF) is estimated using the Nelson–Aalen estimator:

$$\hat{H}(t) = \sum_{j:t_j \leq t} \frac{d_j}{Y_j},$$

where $d_j$ is the number of events at time $t_j$ and $Y_j$ is the number at risk just prior to $t_j$.

**Ensemble Aggregation.** The RSF ensemble survival estimate for an individual is obtained by averaging the node-level CHFs across all trees:

$$\hat{H}_{\mathrm{RSF}}(t \mid x) = \frac{1}{N_T} \sum_{b=1}^{N_T} \hat{H}_b(t \mid x),$$

where $N_T$ is the number of trees. The survival function is then

$$\hat{S}_{\mathrm{RSF}}(t \mid x) = \exp\left(-\hat{H}_{\mathrm{RSF}}(t \mid x)\right).$$

**Key Properties.**

- RSF does not assume proportional hazards or any specific parametric form of survival distribution.
- It can naturally handle high-dimensional covariates, mixed data types (continuous, categorical), and complex nonlinear interactions.
- Variable importance measures and partial dependence plots can provide interpretable insights.

**Limitations.** RSF requires sufficiently large sample sizes for stable performance and may be computationally intensive for large-scale data. Unlike Cox models, RSF does not provide explicit hazard ratios, which can make direct clinical interpretation more challenging.

**Applications.**   RSF has been successfully applied in biomedical research, reliability analysis, and large-scale survival prediction tasks, particularly where nonlinear effects and high-dimensional predictors are present.

### A.3   Simulation of Survival Data Using Copulas

To study dependent censoring, we generate synthetic survival datasets where the joint dependence between latent event times $E$ and censoring times $C$ is introduced via copula functions. A copula $C_\theta(u, v)$ is a bivariate distribution function on $[0, 1]^2$ with uniform marginals, which allows us to couple any two marginal survival distributions into a valid joint distribution with a controlled dependence structure.

The simulation process involves two steps:

1. Sample a pair of dependent uniform random variables $(U, V)$ from a chosen copula $C_\theta$.
2. Transform these uniform variables into event and censoring times using their respective quantile functions (inverse transform sampling), i.e., $E = F_E^{-1}(U)$ and $C = F_C^{-1}(V)$.

We employ four standard parametric copula families. Below, we present the theoretical definition and the exact, well-established simulation algorithm used for each.

**Gaussian Copula.**   The Gaussian copula is defined by the CDF of a bivariate standard normal distribution with correlation coefficient $\rho$.

$$C_\rho(u, v) = \Phi_\rho(\Phi^{-1}(u), \Phi^{-1}(v)), \quad \rho \in (-1, 1).$$

*Simulation Algorithm:* We generate a vector $(Z_1, Z_2)$ from a bivariate standard normal distribution with correlation $\rho$ (typically via Cholesky decomposition of the correlation matrix) and then set $U = \Phi(Z_1)$ and $V = \Phi(Z_2)$.

**Clayton Copula.**   The Clayton copula is an Archimedean copula useful for modeling strong lower-tail dependence (i.e., early events are strongly associated with early censoring).

$$C_\theta(u, v) = \left[\max\{u^{-\theta} + v^{-\theta} - 1, 0\}\right]^{-1/\theta}, \quad \theta \in (0, \infty).$$

*Simulation Algorithm (Marshall-Olkin):* Generate a shared random variable $W \sim \mathrm{Gamma}(1/\theta, 1)$ and two independent variables $E_1, E_2 \sim \mathrm{Exp}(1)$. The dependent uniforms are then $U = (1 + E_1/W)^{-1/\theta}$ and $V = (1 + E_2/W)^{-1/\theta}$.

**Frank Copula.**   The Frank copula is another Archimedean copula that allows for both positive and negative dependence and exhibits symmetric tail dependence.

$$C_\theta(u, v) = -\frac{1}{\theta} \log \left( 1 + \frac{(\exp(-\theta u) - 1)(\exp(-\theta v) - 1)}{\exp(-\theta) - 1} \right), \quad \theta \in \mathbb{R} \setminus \{0\}.$$

*Simulation Algorithm:* This exact algorithm uses a Log-Series distributed variable. Generate $W \sim \mathrm{LogSeries}(1 - \exp(-\theta))$ and two independent variables $E_1, E_2 \sim \mathrm{Exp}(1)$. The dependent uniforms are then $U = 1 - \exp(-E_1/W)$ and $V = 1 - \exp(-E_2/W)$.

**Gumbel Copula.**   The Gumbel copula is an Archimedean copula that models strong upper-tail dependence (i.e., late events are strongly associated with late censoring).

$$C_\theta(u, v) = \exp\left[ - \left((- \log u)^\theta + (-\log v)^\theta\right)^{1/\theta} \right], \quad \theta \in [1, \infty).$$

*Simulation Algorithm:* The standard algorithm requires sampling from a positive stable distribution. We generate a variable $S$ from a stable distribution with characteristic exponent $1/\theta$, skewness 1, scale $(\cos(\pi/(2\theta)))^\theta$, and location 0. Then, given two independent uniform variables $U_1, U_2 \sim \mathrm{Uniform}(0, 1)$, the dependent uniforms are $U = \exp(-(- \log U_1)/S)$ and $V = \exp(-(- \log U_2)/S)$. Our implementation relies on standard library functions for this procedure.

**Fidelity of Simulation.** While the theoretical CDFs are essential for mathematical analysis, the simulation algorithms we employ are the standard, exact methods for generating random variates from these distributions. For example, the Marshall-Olkin and Log-Series constructions are known to yield random variables with distributions that are mathematically identical to the Clayton and Frank copulas, respectively. This ensures that our synthetic data faithfully represents the desired dependence structures, providing a valid basis for evaluating our test's performance.

**Data Generation via a Shared Frailty Model.** This model induces dependence via an unobserved latent variable, or "frailty" $Z_i \sim \mathcal{N}(0,1)$, that simultaneously influences the event and censoring risks for each subject $i$. Given covariates $\mathbf{X}_i$, the proportional hazards for the event time $E_i$ and censoring time $C_i$ are:

$$\lambda_E(t|\mathbf{X}_i, Z_i) = \lambda_{0,E} \exp(\mathbf{X}_i \boldsymbol{\beta} + \alpha_E Z_i)$$
$$\lambda_C(t|\mathbf{X}_i, Z_i) = \lambda_{0,C} \exp(\mathbf{X}_i \boldsymbol{\gamma} + \alpha_C Z_i)$$

The latent times $E_i$ and $C_i$ are sampled from exponential distributions with these rates, and the observed data are $T_i = \min(E_i, C_i)$ and $\delta_i = \mathbf{1}\{E_i \leq C_i\}$. The censoring mechanism is controlled by the frailty coefficients: setting $\alpha_C = 0$ ensures conditionally independent censoring ($H_0$), as the frailty no longer links the two processes. Conversely, setting both $\alpha_E, \alpha_C \neq 0$ (typically with the same sign) induces dependent censoring ($H_1$), where the magnitude of the product $\alpha_E \alpha_C$ governs the dependence strength. The baseline hazard $\lambda_{0,C}$ is adjusted to achieve a target overall censoring rate.

---

**Algorithm 1** Null Distribution Generation via Non-Parametric Imputation

---

**Require:** Original data $\{(T_i, \delta_i, \mathbf{X}_i)\}_{i=1}^n$.
**Ensure:** A single null dataset $\{(T_i^*, \delta_i^*)\}_{i=1}^n$.
    **1. Non-Parametric Modeling:**
1: Train an event model $M_E$ (RSF) on the data to learn $\hat{S}_E(t|\mathbf{X}) = \hat{P}(E > t \mid \mathbf{X})$.
2: Train a censoring model $M_C$ (RSF) using an inverted indicator $\delta' = 1 - \delta$ to learn $\hat{S}_C(t|\mathbf{X}) = \hat{P}(C > t \mid \mathbf{X})$.
    **2. Conditional Imputation of Latent Times:**
3: Initialize empty latent time vectors $\mathbf{E}_{\text{full}}, \mathbf{C}_{\text{full}}$.
4: **for** each subject $i = 1, \ldots, n$ **do**
5:     **if** $\delta_i = 1$ **then**                         ▷ Subject experienced an event
6:         $E_i \leftarrow T_i$.
7:         Sample censoring time $C_i \sim P(C > t \mid C > T_i, \mathbf{X}_i)$ using model $M_C$.
8:     **else**                                         ▷ Subject was censored
9:         $C_i \leftarrow T_i$.
10:        Sample event time $E_i \sim P(E > t \mid E > T_i, \mathbf{X}_i)$ using model $M_E$.
11:     **end if**
12:     Add $E_i$ to $\mathbf{E}_{\text{full}}$ and $C_i$ to $\mathbf{C}_{\text{full}}$.
13: **end for**
    **3. Stratified Permutation:**
14: Create $\mathbf{C}_{\text{perm}}$ by randomly shuffling the elements of $\mathbf{C}_{\text{full}}$ within each stratum defined by the unique values of covariates $\mathbf{X}$.
    **4. Null Dataset Reconstruction:**
15: **for** each subject $i = 1, \ldots, n$ **do**
16:     $T_i^* \leftarrow \min(E_i, C_i^{\text{perm}})$.
17:     $\delta_i^* \leftarrow \mathbf{1}\{E_i \leq C_i^{\text{perm}}\}$.
18: **end for**
19: **return** The null dataset $\{(T_i^*, \delta_i^*)\}_{i=1}^n$.

---

## A.4 The Use of Large Language Models (LLMs)

We used LLMs for polishing the text.

---

**Algorithm 2** Complete Dependent Censoring Test via Non-Parametric Imputation

---

**Require:** Data $\{(T_i, \delta_i, \mathbf{X}_i)\}_{i=1}^{n}$, time points $\{t_k\}_{k=1}^{K}$, number of permutations $B$.

**Part 1: Compute Observed Statistics**
1: **for all** stratum $s$ **do**
2:      Compute observed test statistics: $\Delta I_{t_k}^s$ for $k = 1, \ldots, K$ using the original data.
3:      Let $\Lambda_{\text{obs}}^s \leftarrow \max_k \Delta I_{t_k}^s$.
4: **end for**

**Part 2: Generate Permutation Distribution**
5: Fit RSF models $M_E$ and $M_C$ on the full dataset as described in Algorithm 1.
6: **for** $b = 1, \ldots, B$ **do**
7:      Generate a null dataset $\{(T_i^{*(b)}, \delta_i^{*(b)})\}$ using Algorithm 1.
8:      **for all** stratum $s$ **do**
9:          On the $b$-th null dataset, compute $\Lambda_{\text{perm}}^{s,(b)} \leftarrow \max_k \Delta I_{t_k}^{s,(b)}$.
10:      **end for**
11: **end for**

**Part 3: Aggregate with Fisher's Method**
12: **for all** stratum $s$ **do**
13:      Calculate stratum p-value: $p_s \leftarrow \frac{1 + \sum_{b=1}^{B} \mathbf{1}\{\Lambda_{\text{perm}}^{s,(b)} \geq \Lambda_{\text{obs}}^s\}}{B+1}$.
14: **end for**
15: Calculate observed Fisher statistic: $F_{\text{obs}} \leftarrow -2 \sum_s \log(p_s)$.

16: **for** $b = 1, \ldots, B$ **do**
17:      **for all** stratum $s$ **do**
18:          Compute null p-value: $p_s^{(b)} \leftarrow \frac{1 + \sum_{j=1}^{B} \mathbf{1}\{\Lambda_{\text{perm}}^{s,(j)} \geq \Lambda_{\text{perm}}^{s,(b)}\}}{B+1}$.
19:      **end for**
20:      Calculate null Fisher statistic: $F^{(b)} \leftarrow -2 \sum_s \log(p_s^{(b)})$.
21: **end for**

**Part 4: Calculate Final Global p-value**
22: $p_{\text{global}} \leftarrow \frac{1 + \sum_{b=1}^{B} \mathbf{1}\{F^{(b)} \geq F_{\text{obs}}\}}{B+1}$.
23: **return** $p_{\text{global}}$.

---

