# OpenReview forum: "The Mutual Information Uncertainty Range: A Non-Parametric Test for Dependent Censoring"
_ICLR.cc/2026/Conference — Submitted to ICLR 2026_

### Official Review · Reviewer_7e6M · 2025-10-20

**Soundness:** 3
**Presentation:** 3
**Contribution:** 2
**Rating:** 2
**Confidence:** 2

**Summary:**

The paper proposes a non-parametric diagnostic to test whether event and censoring times are conditionally independent given covariates (E \perp C \mid X). It discretizes time through binary indicators E_t = 1(E > t) and C_t = 1(C > t), and shows that continuous-time independence implies discrete-time independence for each t. Because censoring induces missing cells in the (E_t, C_t \mid X) contingency tables, the authors bound the conditional mutual information over all valid imputations to obtain an uncertainty range. A stratified permutation test—using an RSF-based “impute-and-permute’’ null generation and Fisher’s method to aggregate results—yields a global p-value. Experiments on synthetic data show well-calibrated type-I error and sensitivity to dependence strength.

**Strengths:**

•	Well-posed reduction from continuous to discrete testing; Proposition 1 provides a usable contrapositive for rejection.
	•	Model-free: no parametric assumptions on hazards or dependence; operates directly on counts and censoring patterns.
	•	Clear and transparent testing pipeline, with detailed algorithms for the ΔI statistic and Fisher aggregation.
	•	Synthetic evaluation spans multiple dependence structures.

**Weaknesses:**

• The work is fundamentally a statistical diagnostic, not a machine learning contribution — it introduces no learning algorithm, representation, or model.
	• The “RSF-based imputation” component uses a standard survival model purely as a simulation tool and does not constitute a methodological advance in ML.
	• All experiments are synthetic, with no real or large-scale data to demonstrate practical impact or relevance to modern ML pipelines.
	• The test outputs p-values rather than learned models or predictive improvements, limiting its applicability within the ICLR context.

**Questions:**

1.	How could this diagnostic be integrated into modern machine learning survival pipelines (e.g., DeepSurv, RSF, or neural Cox models) to improve model selection or training?
	2.	Is there a path toward making the proposed test differentiable or learnable, enabling its use as a regularizer or component within end-to-end ML models?

---

### Official Review · Reviewer_WGY5 · 2025-10-27

**Soundness:** 2
**Presentation:** 3
**Contribution:** 2
**Rating:** 2
**Confidence:** 4

**Summary:**

This study develops a statistical test to assess the common assumption of non-informative, also called independent censoring. This assumption is deemed untestable, as the event time (E) and the censoring time (C) are never observed jointly for an individual. The authors propose to test for the underlying structure in the discretized missingness pattern. The authors provide experiments on simulated data with a variety of dependence patterns.

**Strengths:**

The authors address an interesting and underexplored problem. The proposed solution seems adequate and tractable, which is very appealing .The paper is overall clear and well written.

**Weaknesses:**

This work has a number of weaknesses
- The main weakness of this work is that it is mostly empirical, with no theory supporting the sensitivity or other properties of the test,
- the experiences are very limited, with simulated datasets with only four covariates, either all binary, or all continuous, which is rather simplistic.
- The claim that those simulated results "demonstrat[e]  proper error control" based on the fact that for the independent setting, the test returns a large p-value is a little strong compared to the level of evidence.
- There is a double step of discretization (the time, and the covariates, without proper discussion, and empirical demonstration (for the time for example, what is the impact if there happens to be a big gap between the event times?

**Questions:**

- there is a strong need of more realistic, and more thoroug experimental setting, as the work is mostly empirical, with the use of real data, both with semi-simulation (to have a realistic covariate distribution, but the truth regarding the tested hypothesis), and real datasets, to see in practice how the test behaves: is the independent censoring assumption never true (according to the test)? It could also allow some evaluation about the practical usability in terms of dataset size, number of covariates, sensitivity of the results to the hyperparameter of the number of bins, the runtime of the algorithm depending on those data characteristics.
- will the code with proper documentation will be made avaiable? it is absolutely key to the reproducibility of the results, and important for further use of the test

---

### Official Review · Reviewer_RnfC · 2025-10-30

**Soundness:** 2
**Presentation:** 3
**Contribution:** 2
**Rating:** 4
**Confidence:** 4

**Summary:**

This paper introduces a statistical test to evaluate the common assumption in survival analysis. The assumption states that the censoring time and the event time are conditionally independent given the covariates. However, this assumption is usually not tested (on synthetic datasets or on real-life datasets). Previous tests relied on additional assumptions (such as proportional hazards), and this work represents a novelty in overcoming that limitation.

**Strengths:**

This paper has several strengths:
- The authors introduce a new statistical test to assess the common assumption E \indep C | x in survival analysis. This test does not rely on additional assumptions (unlike previous statistical tests), which makes this work of high quality.
- The theoretical aspect of the paper is well written, with clear proofs and a lot of intuitions.
- The process for generating independent event and censoring times using RSF is very interesting.

**Weaknesses:**

Major weaknesses:
- In Section 4.1, the authors could provide more information about the CMI.
- Although the authors mention that several existing tests are available, they did not compare their proposed test with these alternatives. It would be interesting to see how the other tests perform compared to theirs, and under which settings.
- Some experiments could also have been conducted on real-life datasets, for instance those available in the python library pycox, to evaluate whether the proposed test passes in practical situations.
- The experimental tables lack clarity. The authors could use colors or redesign the tables to better visualize independence (or dependence) through the p-values across different settings.

Minor weaknesses:
- The paper contains several typos, e.g. line 78: eahch -> each; line 159: xor -> or; line 247: missing parenthesis.

**Questions:**

The questions I have are related to the weaknesses:

- Could you explain a bit more about the creation of the synthetic data using RSF? Does the distribution of the generated datasets resemble that of other survival analysis datasets? And what is the distribution of $\Delta I$ when the assumption holds?

- Would it be possible for you to add the following three tests: the Cox-based test (Lee & Wolfe), the Rizopoulos test based on copulas, and the Sun & Lee test? This could really strengthen your paper.

- Also, could you include the results for the real-life datasets (from the Python library pycox or scikit-survival) for both your test and the previous ones? It would help users understand how many tests have passed before using these datasets and methods that assume  E \indep C | x

---

### Official Review · Reviewer_2WPk · 2025-11-01

**Soundness:** 2
**Presentation:** 2
**Contribution:** 2
**Rating:** 4
**Confidence:** 2

**Summary:**

This paper aims to design a nonparametric test for diagnosing conditional independent censoring $(E \perp C \mid X)$ in survival data. Continuous time is discretized and binary event/censoring indicators with ``?'' for unobserved entries form partially observed contingency tables per stratum. The test statistic is the conditional mutual information (CMI) uncertainty range $\Delta I_t$, defined as the gap between upper and lower CMI over all valid completions of the tables. These bounds are solved via a decomposable integer program with dynamic programming, and significance is assessed using a stratified permutation procedure.

Synthetic experiments with Gaussian, Clayton, Gumbel, and Frank copulas, plus a frailty model, show Type-I error control near independence and increasing power as dependence strengthens.

**Strengths:**

1. Problem relevance: Targets the untestable yet ubiquitous independent-censoring assumption, which is very commonly used in survival analysis.

2. Optimization setup: bounding CMI by integer variables per stratum is natural.

**Weaknesses:**

1. “Assumption-free” is overstated: the null depends on RSF models and on discretization choices for continuous X. Sensitivity to bin counts and RSF hyperparameters is not studied.

2. No stress tests with small strata, heavy censoring, ties, or time-dependent covariates.

3. No power analysis beyond plots; no characterization of when bounds are tight.

**Questions:**

1. How sensitive is Type-I control to RSF mis-specification, and would a conditional-randomization design reduce model dependence?

2. Could you provide some stress tests with small strata, heavy censoring?

---

### Official Review · Reviewer_fYPT · 2025-11-01

**Soundness:** 3
**Presentation:** 3
**Contribution:** 3
**Rating:** 6
**Confidence:** 1

**Summary:**

The paper studies the problem of testing if dependent censoring exists in a dataset. The authors propose a statistical test based on *conditional mutual independence*, by comparing it against the null hypothesis; if the CMI is too large or small compared to the null hypothesis then the null hypothesis that there is independent censoring is rejected.

The overall method is based on the following contributions (i) showing that if independent censoring holds, then the same holds in the discrete case (hence allowing the authors to focus on discrete time settings thereafter for computational reasons), (ii) introducing CMI uncertainty range as a test statistic of sorts, (iii) proposing a more efficient method to compute CMI based on separability, and (iv) proposing a way to estimate the CMI under the null hypothesis.

Note: this is outside my reviewing expertise. I have brought this up to the AC and defer to other reviewers.

**Strengths:**

- The paper is easy to understand and follow, even for someone outside the field. The contribution and motivation is clear

**Weaknesses:**

- I do not see any obvious big weaknesses, but the empirical evaluation does seem a little limited. The Clayton, Gumbel, and Frank copula are all Archimedean which are quite restrictive. I wonder if there are richer joint survival distributions that could be used to stress test the proposed method.

**Questions:**

- Can the authors comment on whether imputation is an acceptable range of computing CMI? My intuition is that it may be that the difference between min and max would be so large that the test becomes too weak to be useful.

---

### Meta-Review · Area_Chair_pjKt · 2025-12-30

**Summary:**

Across reviews, there is agreement that the paper tackles an important and practically relevant issue in survival analysis: diagnosing violations of conditional independent censoring. Reviewers recognize clear exposition and a technically interesting core—computing conditional mutual information (CMI) bounds over partially observed contingency tables via a decomposable integer program / dynamic program—and synthetic results that suggest calibration and increasing power with stronger dependence (fYPT, 7e6M). The main concerns driving the recommendation are (i) limited empirical validation (all-synthetic; limited stress tests), (ii) unclear robustness to discretization and the RSF-based null-generation procedure, (iii) lack of comparisons to existing dependent-censoring tests, and (iv) questions about fit/impact for ICLR given the diagnostic/statistics framing rather than an ML method (2WPk, RnfC, WGY5, 7e6M).

**Reviewer Concerns:**

The rebuttal appears to have helped clarify the intended scope (a diagnostic rather than a learning method) and addressed several clarification requests about the pipeline and motivation, partially reducing confusion about what is “assumption-free” versus what depends on discretization and the null-generation mechanism. However, multiple substantive requests remain outstanding: systematic sensitivity analyses (binning/discretization choices, RSF hyperparameters/mis-specification), stronger stress tests (small strata, heavy censoring, ties, time-dependent covariates), comparisons to established tests in the literature, and at least a minimal real-data evaluation to demonstrate practical value (2WPk, RnfC, WGY5). In addition, the venue-fit concern—how this diagnostic concretely advances or plugs into modern ML survival pipelines—remains only partially resolved (7e6M).

**Reviewer Scores:**

fYPT: Likely unchanged around 6; the review was positive but explicitly low-confidence/out-of-area, so rebuttal would not strongly move the score.

2WPk: Potentially 4 → 4/6 if rebuttal convincingly narrows “assumption-free” claims and explains robustness; still likely below/around threshold without new sensitivity experiments.

RnfC: Potentially 4 → 4 the key gap (baselines + real data) likely keeps it unchanged.

WGY5: Likely 2 → 2/4 at best; concerns focus on lack of theory and limited/idealized experiments, which are hard to fully address without additional results.

7e6M: Likely 2 → 2/4 depending on how persuasively the rebuttal argues ICLR relevance/integration; absent empirical evidence of ML pipeline impact, likely stays negative.

---

### Decision · Program_Chairs · 2026-01-26

Reject